# Extreme temperature events on Greenland in observations and the MAR regional climate model

Amber A. Leeson[1], Emma Eastoe[2], Xavier Fettweis[3]

[1]Lancaster Environment Centre / Data Science Institute, Lancaster University, Lancaster, LA1 4YW, UK.
[2]Department of Mathematics and Statistics, Lancaster University, Lancaster, LA1 4YW, UK.
[3]Department of Geography, University of Liege, 4000 Liège, BELGIUM

*Correspondence to*: Amber A. Leeson (a.leeson@lancaster.ac.uk)

**Abstract.**

Melt water from the Greenland ice sheet contributed 1.7-6.12 mm to global sea level between 1993 and 2010 and is expected
to contribute 20–110 mm to future sea level rise by 2100. These estimates were produced by regional climate models which
are known to be robust at the ice-sheet scale, but occasionally miss regional and local scale climate variability (e.g. Leeson et
al., 2017, Medley et al., 2013). To date, the fidelity of these models in the context of short period variability in time (i.e. intra-
seasonal) has not been fully assessed, for example their ability to simulate extreme temperature events. We use an event
identification algorithm commonly used in Extreme Value Analysis, together with observations from the Greenland Climate
Network (GC-Net), to assess the ability of the MAR RCM to reproduce observed extreme positive temperature events at 14
sites around Greenland. We find that MAR is able to accurately simulate the frequency and duration of these events but
underestimates their magnitude by more than half a degree Celsius/Kelvin, although this bias is much smaller than that
exhibited by coarse-scale Era-Interim reanalysis data. As a result, melt energy in MAR output is underestimated by between
16% and 41% depending on global forcing applied. Further work is needed to precisely determine the drivers of extreme
temperature events, and why the model underperforms in this area, but our findings suggest that biases are passed into MAR
from boundary forcing data. This is important because these forcings are common between RCMs and their range of predictions
of past and future ice sheet melting. We propose that examining extreme events should become a routine part of global and
regional climate model evaluation and addressing shortcomings in this area should be a priority for model development.

## 1 Introduction

Since the 1990s, the Greenland Ice Sheet has shifted from a state of near mass balance, to one of significant mass loss (Shepherd
et al., 2012, Hanna et al., 2013a, van den Broeke et al., 2016), contributing approximately 10% to the measured global sea
level rise during the last two decades (Church, 2013). Since 2010, the rate of mass loss from Greenland has increased and the
ice sheet has experienced episodes of rare and extreme surface melt (Nghiem et al., 2012, Hanna et al., 2014, Tedesco et al.,
2013). For example in 2012, the summer melt extent reached 98.6% of the entire ice sheet; thought to be the greatest melt
extent in over a century (Nghiem et al., 2012). In addition to directly removing more of the ice sheet into the sea, melting

reduces the reflectivity of the ice sheet and can warm the perennial snow pack (through latent heat release when the melt water refreezes), both of which act as a positive feedback to further enhance melt. These processes also alter the dielectric properties of the ice sheet surface, which makes it more difficult to measure surface height change using satellite-borne radar instruments (McMillan et al., 2016). An understanding of the location, frequency, duration and magnitude of melting is therefore necessary
to 1) understand the ice sheet's response to climate change, 2) interpret contemporary measurements of ice sheet volume change and 3) to constrain predictions of future ice sheet state.

Mass lost through meltwater runoff and gained through snowfall together comprise the ice sheet's surface mass balance (SMB), which is typically assessed at the ice sheet wide scale using Regional Climate Models (RCM). RCMs act as physically based
interpolators of relatively coarse resolution climate reanalysis data, and produce high resolution estimates in areas where the local climate exhibits high spatial variability i.e. ice sheet margins (Noel et al., 2016). Alternative statistical downscaling techniques fulfil a similar purpose and give broadly comparable results (Wilton et al. 2017, Vernon et al. 2013). RCMs can also make high resolution predictions of future climate, when boundary forcing is applied by global climate model (GCM) output instead of reanalysis data. In the last IPCC report, the MAR, RACMO2 and MM5 RCMs reported that while SMB
remains positive (net increase in mass due to surface processes) increases in melting were responsible for a sea level contribution of 0.23-0.64 mm yr$^{-1}$ during 2005-2010 (Church, 2013). RCMs are known to perform well when compared to integrated quantities, for example mean annual melt measured at weather stations or total mass loss from the ice sheet measured by GRACE (van den Broeke et al., 2016). However, fidelity at the regional or seasonal scales does not necessarily translate to the local scale (e.g. Medley et al., 2013). Extreme melt events, for example, tend to be localised in time (typically only lasting
for a day or so). While RCM predictions of melt *extent* during extreme events have been found to be reliable (Tedesco et al., 2011), an assessment of their ability to simulate the frequency, duration and magnitude of these events, and how this might affect their projections of future ice sheet change, has yet to be performed.

In this paper, we use advanced statistical techniques for extreme event identification to compile a statistical climatology of
extreme temperature events on Greenland since the 1990s using data from 14 automatic weather stations forming part of the Greenland Climate Network (GC-Net, Steffen et al., 1996). Note that these are distinct from extreme melt years as it is possible to have multiple extreme temperature events in a year. We then use these data, together with temperature estimates from the MAR regional climate model (Fettweis et al., 2017) to evaluate the model's ability to capture the frequency, duration and magnitude of these events when forced by climate reanalysis and by GCM data. Finally, we estimate melt energy available at
the GC-Net stations during this time using a positive degree day sum (PDD) and assess the degree to which discrepancies between observed and modelled characteristics of extreme events affects MAR based estimates of melt energy.

## 2 Methods and Data

### 2.1 Greenland Climate Network data

The Greenland climate network (GC-Net) consists of 18 automatic weather stations (AWS) distributed around the ice sheet. We refer the reader to Steffen et al. (1996) for details but briefly summarise here. The first station (Summit) began operation in 1995, with others coming online at various times since then. The AWS measure a range of meteorological variables, of which the temperature and pressure time series are the most complete. The GC-Net stations each have four temperature sensors (2 different instruments mounted at 2 different heights), here we use data from the Type-E Thermocouple mounted at position 1 at all sites except NGRIP, and Saddle during 2010-2016, for which we used data from the Type-E Thermocouple mounted at position 2. Measurements are taken hourly, we use these data to calculate daily maximum and mean values for compatibility with MAR output.

Our analysis focuses on 14 of the 18 stations; we found the remaining 4 stations to have temperature time series which were either too short or too patchy for robust statistical analysis. Figure 1 shows the data coverage at each of the 14 stations studied here and Table 1 gives the total number of years of data available, when gaps are excluded. We attribute these missing data to equipment failure and assume that it is unrelated to the occurrence of extreme high temperatures. As such we treat these data gaps as `missing at random' and ignore them in our analysis. Since most of the missing periods cover whole years, rather than just a summer- or winter-period, this assumption is reasonable.

### 2.2 MAR Regional Climate Model

The MAR model is an RCM developed and extensively evaluated to study the present Greenland climate and SMB from the beginning of the last century (Fettweis et al., 2017) as well as to perform future projections of Greenland Ice Sheet SMB for the last IPCC report till the end of this century (Fettweis et al., 2013). It is fully coupled with a snow energy balance model dealing with the energy and mass exchanges between surface, snow, ice and atmosphere. The MAR version 3.5 used here has been extensively evaluated in Fettweis et al. (2017) with daily in situ PROMICE based AWS measurements over 2008-2010, daily satellite derived melt extents over 1979-2010 as well as SMB measurements and ice cores over 1958-2010. We chose to use MARv3.5 in this study since this is the model version which was used to make the most recent set of estimates of future ice sheet change (Fettweis et al., 2013). We refer to Fettweis (2007) and Fettweis et al. (2013, 2017) for more details about MAR and its surface scheme.

Here, we use data from MAR simulations forced with the ERA-Interim reanalysis (e.g. Fettweis et al., 2017), and with the GCMs CanESM2, MIROC5 and NorESM1 over 1995-2015. CanESM2, MIROC5 and NorESM1 have been found to be the best models (in respect to ERA-Interim over 1980-1999) from the CMIP5 database over Greenland from which 6 hourly outputs were available (Fettweis et al., 2013). MAR is forced every 6 hours at its lateral boundaries with temperature, humidity,

wind and surface pressure. Sea surface temperature and sea ice extent is also prescribed into the MAR integration domain from the forcing data every 6 hours. Hereafter we refer to MAR variants with forcing by Era-Interim, NorESM1, CanESM2 and MIROC5 as MAR-Era, MAR-Nor, MAR-Can MAR-MIR, respectively. MAR-Era data are available continuously during our study period (1995-2015 inclusive). For the GCM forced model runs, we use historical simulations until 2006 and simulations performed under forcing by the RCP8.5 climate change scenario (van Vuuren et al., 2011) thereafter. This is reasonable because observed greenhouse gas concentrations followed the RCP8.5 scenario during this period, and in any case the differences between the RCP scenarios during 2006-2015 are very small. For comparison with the GC-Net data we pick the MAR model grid cell (25 km by 25 km resolution) closest to the AWS location in terms of latitude and longitude of the cell centre. The MAR cell centre is typically at a lower elevation than the AWS, according to the MAR DEM and the measured elevation of the AWS, and so we apply a lapse rate based correction to MAR temperature data ($0.71^{\circ}$C per 100 m of elevation difference, Steffen and Box (2001)). We restrict the model time series at each station to periods where GC-Net data are also available.

## 2.3 Extreme Value Analysis (EVA)

Extreme value analysis provides a toolbox of methods for the identification and statistical modelling of extreme events (Coles, 2013) i.e. events that are unusually large or small when compared to the central behaviour of a dataset. For a given site and a given data type (observations, MAR-Era, MAR-MIR, MAR-Nor and MAR-Can), we identify the extreme events using a site- and type-specific threshold applied to the maximum daily temperature time series. To enable a fair comparison, the threshold is taken always to be the 90% quantile of the dataset in question (Table 2) and an extreme event is deemed to start once the maximum daily temperature exceeds this threshold. The event ends after the temperature has been below the threshold for three consecutive days. This method of event identification is known as the runs method (Smith and Weissman, 1994). It follows that the durations, as well as both frequencies and magnitudes, of events are random. Note that here we take the magnitude of an event to be the largest of the daily maxima within that event.

## 2.3 Positive Degree-Day Sum

Melting is most appropriately calculated as a function of the surface energy balance; however measurements of variables required to calculate the surface energy balance (e.g. net radiation, wind speed) are not consistently available at the GC-Net stations. Positive Degree Days are an estimate of the magnitude and duration of above-zero temperature events and are typically well-correlated with melting (e.g. Braithwaite (1995), Huybrechts et al. (1991)). Here we calculate positive degree-days (PDD) for both observed and modelled temperatures and take this to be a reasonable approximation for melt energy. Diurnal temperature variability is modelled using eq 1 and PDDs are calculated by integrating eq 1 where T > $0^{\circ}$C.

$T = A sin(\varphi t) + B$       (1)

Where A is daily maximum temperature, B is daily mean temperature and φ is one day. Daily mean and maximum 2 m temperature are output by MAR; for GC-Net data daily mean and maximum are calculated based on hourly data as detailed above.

## 2.3 Melt zone definitions

We use independent definitions of the ablation, percolation and dry snow zones first identified in McMillan et al. (2016) using
RACMO2.3 simulations of SMB and surface melt. Briefly, the area of the ice sheet lying below the equilibrium line in a majority of years between 2009 and 2014 is defined as the ablation zone. The area of ice where melt did not exceed 5 mm w.e. on any day during this period is defined as the dry snow zone, with the remainder being classed as the percolation zone. Using these definitions we find areas of 0.23, 0.61 and 0.80 million $km^2$ for the ablation, percolation and dry snow zones respectively.

## 3 Results

**3.1 Extreme temperature events**

We apply extreme value analysis to observed daily maximum temperatures from GC-Net in order to compile a statistical climatology of extreme temperature events on Greenland (Table 3). Each location is considered independently and the timing of statistically extreme events is not necessarily contemporaneous between stations. Extreme events are characterised in terms of their frequency, duration and magnitude; we use 'duration' and 'magnitude' to refer to median values across all events
observed/modelled. We assess these characteristics in the context of station geography i.e. elevation, latitude and melt zone (see methods). Each of the three characteristics is dependent on elevation, however the nature of that dependence and the role that latitude and melt zone play in the relationship is different for each (Figure 2). Extreme temperature events occur 4-8 times per year and event frequency is negatively correlated with elevation in South Greenland; events become less frequent the higher the station is on the ice sheet. Event frequency is positively correlated with elevation in North Greenland/the dry snow
zone (Figure 2a). Events last between 5-10 days, and duration is positively correlated with elevation for all stations (Figure 2b). However events tend to last longer (by ~1 day) at stations in the dry snow zone/North Greenland than at stations at similar elevations in the percolation zone/South Greenland. Event magnitude is negatively correlated with elevation at all stations (Figure 2c), but elevation has a stronger influence on event magnitude in the dry snow and ablation zones (-4.4 (+/- 0.3) ℃ $km^{-1}$) than in the percolation zone (-1.8 (0.4) ℃ $km^{-1}$).

We compare the degree to which MAR is able to capture the observed climatology of extreme events at GC-Net stations by repeating the same extreme value analysis with output from each of the MAR model variants (Figure 3). In addition to considering each station independently, we also consider each model variant independently, i.e. there is no common event mask. This is because the GCM forced model variants (MAR-MIR, MAR-Nor and MAR-Can) are designed to simulate
climatic variability over typically climatic periods like 20-30 yrs, which is not necessarily contemporaneous with observed

variability in a given time period. We exclude JAR2 from the remainder of this analysis due to the large discrepancy in elevation between the station and the corresponding grid cell in MAR (316 m, Table 1) and the dependency we find between elevation and extreme event characteristics (Figure 2). Whilst all of the four model variants typically simulate the duration of extreme events reasonably well (i.e. within 1 day per event), they underestimate event frequency at most of the stations (Figure 3). This is most notable for the GCM forced model variants MAR-MIR, MAR-Nor and MAR-Can which underestimate event frequency by 1.12, 0.75 and 0.65 events per year respectively. Similarly, all of the model variants underestimate the observed event magnitude by more than half a degree at most of the stations (though notably not the two remaining in the ablation zone, in fact event magnitude at Swiss Camp is overestimated). In terms of the individual model variants, the MAR-Era simulation is best able to reproduce event frequency (-0.09 events on average), the MAR-ERA and MAR-MIR simulations are both best able to reproduce event duration (+/-0.04 days on average) and the MAR-Era simulation is best able to reproduce event magnitude (-0.76 °C on average). MAR-Nor is the poorest performing model variant overall.

## 3.1 Mean temperature and mean summer temperature at GC-Net stations in MAR

We assess the ability of the four MAR variants to reproduce temperature observed by the GC-Net more generally by comparing the mean and trend of the entire daily mean temperature time series at each station location (Table 4). We present aggregate statistics for the entire time period in order to account for the fact that the GCM forced MAR variants are predicting climatic variability at the decadal scale. The number of years of data (including gaps in the time series) is given for each station in Table 1. Results are presented by melt zone, where values are an average of all stations in that zone, weighted by the number of years of data available for each station. Both MAR-Era and MAR-MIR overestimate mean daily mean temperature (i.e. the average of all of the daily means) by ~1°C, although this signal is dominated by a large discrepancy in the dry snow zone where both model variants are too warm by ≥1.5°C. Both model variants however, show good agreement with the observations in the ablation zone (-0.24°C and -0.34°C, respectively), which is where the most melting occurs. MAR-Can and MAR-Nor both underestimate temperatures overall, and give better agreement with observations in general (-0.13°C and -0.21°C, respectively), but they exhibit a poor performance in the ablation zone (both variants > 1°C too cold). Considering only the summer (JJA) daily mean temperatures, with the exception of MAR-Can in the percolation zone, all MAR variants are too cold in all zones and overall. MAR-Can performs best overall in summer, with a bias of just -0.01°C. All model variants reproduce observed trends in both all and summer temperatures to within 0.1°Cyr$^{-1}$ (most within 0.05°C per year). We evaluate the ability of MAR-Era to reproduce observed climate variability by comparing modelled vs observed mean annual and mean summer (JJA) temperatures (Figure 4). MAR-Era is well able to capture observed inter-annual variability in both. Mean annual temperatures are particularly well correlated with Pearson's correlation co-efficient (r) values in the range 0.77-1.00. Inter-annual variability in mean summer temperatures is less well captured (r = 0.62-94, if JAR2 is ignored). The low bias in summer temperatures described above is also evident at the inter-annual timescale in the MAR-Era simulation.

### 3.3. Extreme temperature events in Era-Interim data

We assess the degree to which the raw Era-Interim output (i.e. not MAR forced with Era-Interim) captures extreme temperature events at GC-Net stations (Figure 5). In comparison with the same data for MAR-Era, using the raw Era-Interim output yields a poorer match to observations at all sites except NASA-U and NGRIP. The average absolute bias in magnitude of extreme temperature events is 0.87°C in MAR-Era and 1.81°C in the raw Era-Interim data output. In general, Era-Interim underestimates temperatures during extreme events in a similar manner to MAR-Era. However Era-Interim overestimates temperatures in the Swiss Camp region and north east of the ice sheet.

### 3.3 Melting during extreme temperature events

We use a positive degree day sum (see methods) to approximate melt energy available during extreme and non-extreme conditions at each of the 13 stations (Figure 6). We note that the difference in abundance of melt energy between adjacent melt zones is roughly an order of magnitude, with observed total PDDs per station of 617, 96 and 5 °C in the ablation, percolation and dry snow zones respectively. All MAR variants are able to reproduce this gradient. We find that the dependence of observed melt energy on statistically extreme temperatures also scales with elevation ($r^2$=0.71, n=13); one-third and ~95% of all melting occurs during extreme events in the ablation and dry snow zones respectively. MAR-Era is able to reproduce this pattern ($r^2$=0.71) but the relationship is less clear in data from the GCM forced variants ($r^2$ = 0.23-0.43).

We compare differences in the total PDDs observed and predicted during the entire study period; all of the MAR variants are found to underestimate total PDDs (Table 5). During extreme events, we see a two-fold increase in the model bias for the MAR-Era, MAR-MIR and MAR-Can model variants; PDDs are underestimated by 26%, 32% and 22% during extreme events, and 12%, 18% and 10% respectively during non-extreme conditions. In the MAR-Nor simulation, PDDs are underestimated to a greater degree during non-extreme temperature conditions. The relative influence of model bias during extreme events is spatially variable. In the ablation zone, the total bias during extreme events is comparable to that during non-extreme events except that the two biases are of the opposite sign; PDDs are over-estimated in the ablation zone during extreme events and under-estimated in the ablation zone during non-extreme events. Given the relative contribution of melting during extreme events to overall melting here however (just 33%), this results in an underestimate overall of 5%, 12%, 22% and 7% for MAR-Era, MAR-MIR, MAR-Nor and MAR-Can respectively. Conversely, in the percolation zone, PDDs are underestimated during extreme events and over-estimated during non-extreme conditions. However again the signal observed during the dominant regime (i.e. 96% of PDDs occur during extremes) leads to a large underestimate overall (52%, 58%, 84% and 40% for the model variants as before). In the dry snow zone it is more difficult to partition the relative influence of extreme vs non extreme events on total PDDs because there is far less melting here; a PDD total of 5 °C per station over the entire study period. This is particularly of note for Summit and NGRIP stations which are high up and far inland on the ice sheet; very small amounts of melting are observed here but no melting is modelled by any of the model variants.

## 4 Discussion

### 4.1 Extreme temperature events in GC-Net Observations

Despite our relatively small sample, given the size of the ice sheet, we see clear relationships between extreme event characteristics and elevation, latitude and melt regime. It is not surprising that extreme temperature events exhibit a stronger magnitude at lower-lying locations, given the atmospheric temperature lapse rate, but it is interesting that this relationship is less strong for the five percolation zone stations than for stations in the ablation and dry snow zones. We speculate that this is a result of heat exchange at the snow surface moderating near-surface temperatures in this region; sublimation is a known energy sink in the percolation zone in the summer (Ettema et al., 2010). In South Greenland, extreme events at lower elevations tend to be more frequent and of shorter duration than those higher up on the ice sheet. Temperature anomalies can be associated with cloudiness (reflecting upwelling longwave radiation back down to the surface) and lower-lying stations are more likely to experience short-term periods of orographic cloud cover. This is particularly likely to affect West Greenland which lies in the path of the prevailing summer circulation pattern and consequently receives moisture-laden onshore flow during the summer (Ohmura and Reeh, 1991). In North Greenland however, we see that extreme events become both longer-lasting, and more frequent, as elevation increases. Longer extreme temperature events are likely associated with high pressure conditions which are relatively persistent. In fact, extreme melt *years* on Greenland have been attributed to an increase in the frequency and duration of high pressure conditions promoted by wider scale atmospheric pressure gradients such as the North Atlantic Oscillation and the Greenland Blocking Index (e.g. Nghiem et al., 2012, Hanna et al., 2013, Lim et al., 2016, Hanna et al., 2016). Extreme temperature events are responsible for the vast majority of melt energy produced in the percolation and dry snow zones on the ice sheet but contribute a much smaller proportion to overall melt energy in the ablation zone. Because we only have data for two ablation zone stations which are located in close proximity, further work is required to assess whether this is a general property of the ablation zone or restricted to this location; temperatures in general are much warmer here, and extreme events are not required to generate melting.

### 4.2 Extreme temperature events in MAR simulations

All of the four MAR model variants underestimate the frequency of extreme events but simulate their duration well. This suggests that MAR is able to reproduce the persistence of conditions driving extreme temperature events when they arise in the model. All MAR variants under-estimate the magnitude of extreme temperature events at most stations, in most cases by >0.5ºC. This can be explained in part by a general low bias in modelled summer temperatures; although the magnitude of this bias is not sufficient to account for the magnitude of the data-model mismatch during extreme periods. For example, MAR-Era exhibits a bias of -0.35ºC during summer and -0.76ºC during extreme temperature events. The raw Era-Interim output also exhibits a low bias during extreme temperature events at most of the GC-Net stations, with notable exceptions being North East Greenland and the most marginal stations at which temperature during extremes is over-estimated. This suggests that the low bias we see in the MAR model during extreme periods could be an artefact of the forcing data. This is important because

Era-Interim and the GCMs examined here are commonly used to force other regional and local scale models (e.g. RACMO2); their use is not restricted to MAR. The version of MAR which is analysed here (v3.5) is known to underestimate the atmospheric liquid water content and so cloudiness (Fettweis et al., 2017) which may also contribute to the cold bias in temperature extremes. However, we repeated the analysis with the most recent version of MAR (v3.7) in which a correction
for this has been incorporated and this yielded no noticeable difference in the result. All of the MAR model variants and Era-Interim over-estimate event magnitude at stations in the ablation zone, JAR and Swiss Camp. We attribute this to difference in albedo between the bare ice in the ablation zone and the snow-covered surface at higher elevations. Energy exchange in bare ice areas is generally more sensitive to sunny conditions; this likely explains why the biases are opposite in this area compared to the percolation and dry snow zones where the albedo is high enough to prevent this sensitivity.


Melt energy simulated by MAR is underestimated by 19%, 25%, 41% and 16% when forcing is provided by Era-Interim, MIROC5, NorESM1 and CanESM2 respectively. However during extreme events, model biases in terms of melt energy are double those calculated during non-extreme, positive temperature, conditions. This is important because approximately half of all melt energy is generated during extreme events. In general, the GCM forced MAR simulations perform more poorly than
the Era-Interim forced simulation, with the exception of MAR-Can (bias = 16% vs 19% for MAR-Era). We would expect the reanalysis forced simulation to perform the best, given its assimilation of observations; however we note that the difference is not large.

We observe melt energy generated at the two highest/furthest inland stations in our sample; Summit and NGRIP, but none of
the MAR variants simulate any melting at either of these stations during our study period. This is because extreme temperatures are underestimated by ~1°C by MAR at these stations (e.g. MAR-Era exhibits a bias of -0.91 at Summit and -0.76 at NGRIP). It is important to note that these are very small quantities and would not impact ice-sheet wide estimates of melting, however melting is also important because of its role in ice sheet albedo; wet snow is less reflective than dry snow. A significant melt event can be defined as achieving > 1 mm WE/day (Franco et al., 2013), and with the exception of Summit in 2012 this was
not achieved at either station during the study period. Nonetheless, as the climate warms melting at these locations is likely to be more abundant and properly capturing temperature variability here will become even more important.

## 4 Conclusions

Analysis of GC-Net temperature data shows that the frequency, magnitude and duration of extreme temperature events on Greenland are strongly controlled by geography (e.g. elevation, latitude etc.), though further work is needed to determine the
relative contributions of potential physical drivers of extreme events at different locations and over different time periods. The MAR regional climate model accurately predicts the duration of extreme temperature events on Greenland but underestimates their frequency by around 1 day per year and underestimates event magnitude by >0.5°C. While this is an improvement over

coarse-scale reanalysis data, it nonetheless leads to an under-estimate in melt energy, which we calculate to be 16-41% during our study period, dependant on model forcing chosen. MAR-based predictions of future melting are calculated using an energy

balance method which has been shown to perform well against observations in the past (Fettweis et al., 2017). However since temperature plays a significant role in the energy balance equations (though melt does not linearly increase with temperature), it is likely that these predictions are affected by the exaggerated model bias we find during extreme events in our study. Further work is needed to determine why the model underperforms in this area, and if other similar models have the same limitation. We identify this as a model development priority to ensure that MAR based estimates of ice sheet change are both

comprehensive and robust.

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

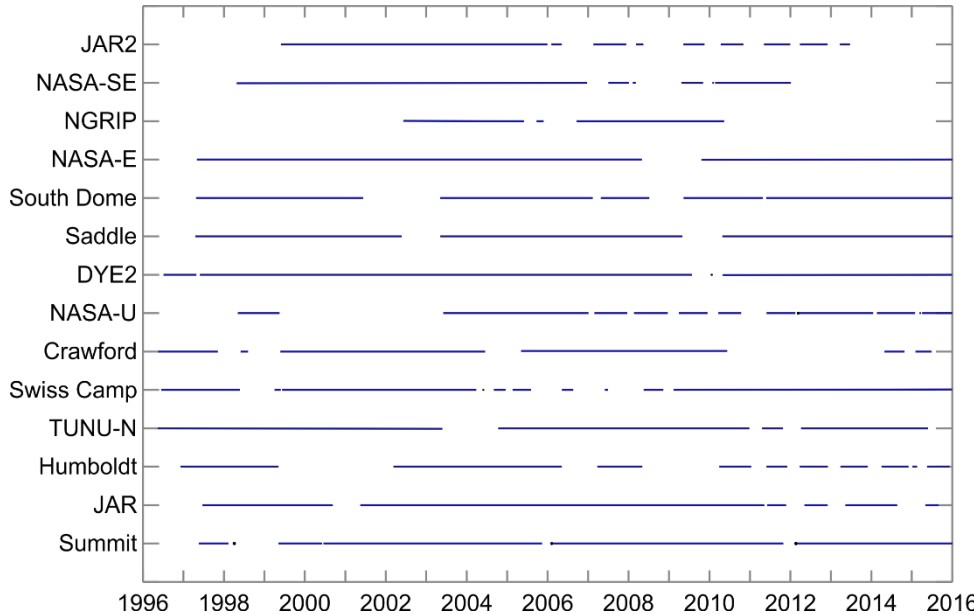

**Figure 1: Data coverage in GC-Net temperature record.**

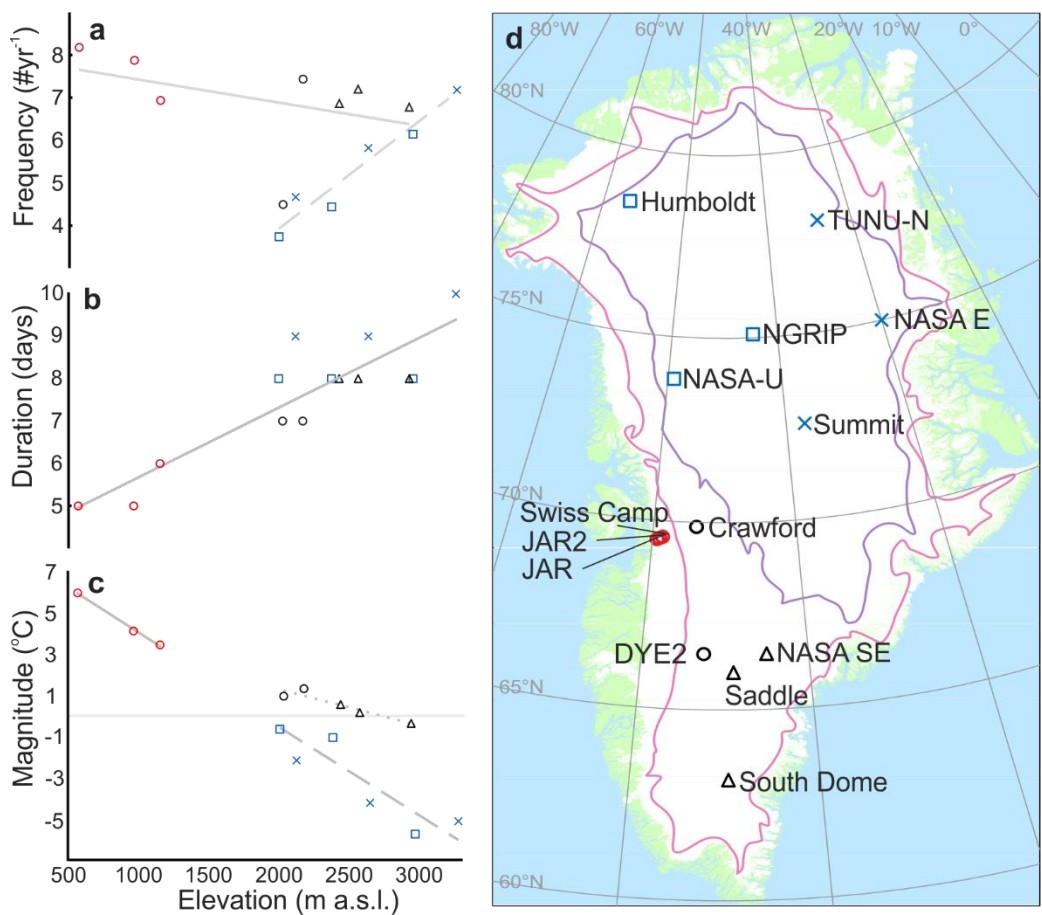

**Figure 2: Frequency, duration and magnitude of extreme events observed at GC-Net automatic weather stations.**
Shapes indicate location: south west = circle, north west = square, north east = cross, south east = triangle. Saddle is
located on the ice divide and we choose to represent it as a south east station. Colours indicate melt zone: red = ablation
zone, black = percolation zone, blue = dry snow zone. In (a) solid grey line represents a linear fit to data from the
ablation and percolation zone stations, dashed line in (a) represents a linear fit to dry snow zone station data. In (b),
solid grey line represents a linear fit to all data. In (c) solid grey line represents a linear fit to ablation zone stations,
dashed grey line represents a linear fit to dry snow zone stations and dotted grey line represents a linear fit to
percolation zone stations. In (d) contours delineate lower limit of percolation (pink) and dry snow (purple) zones.

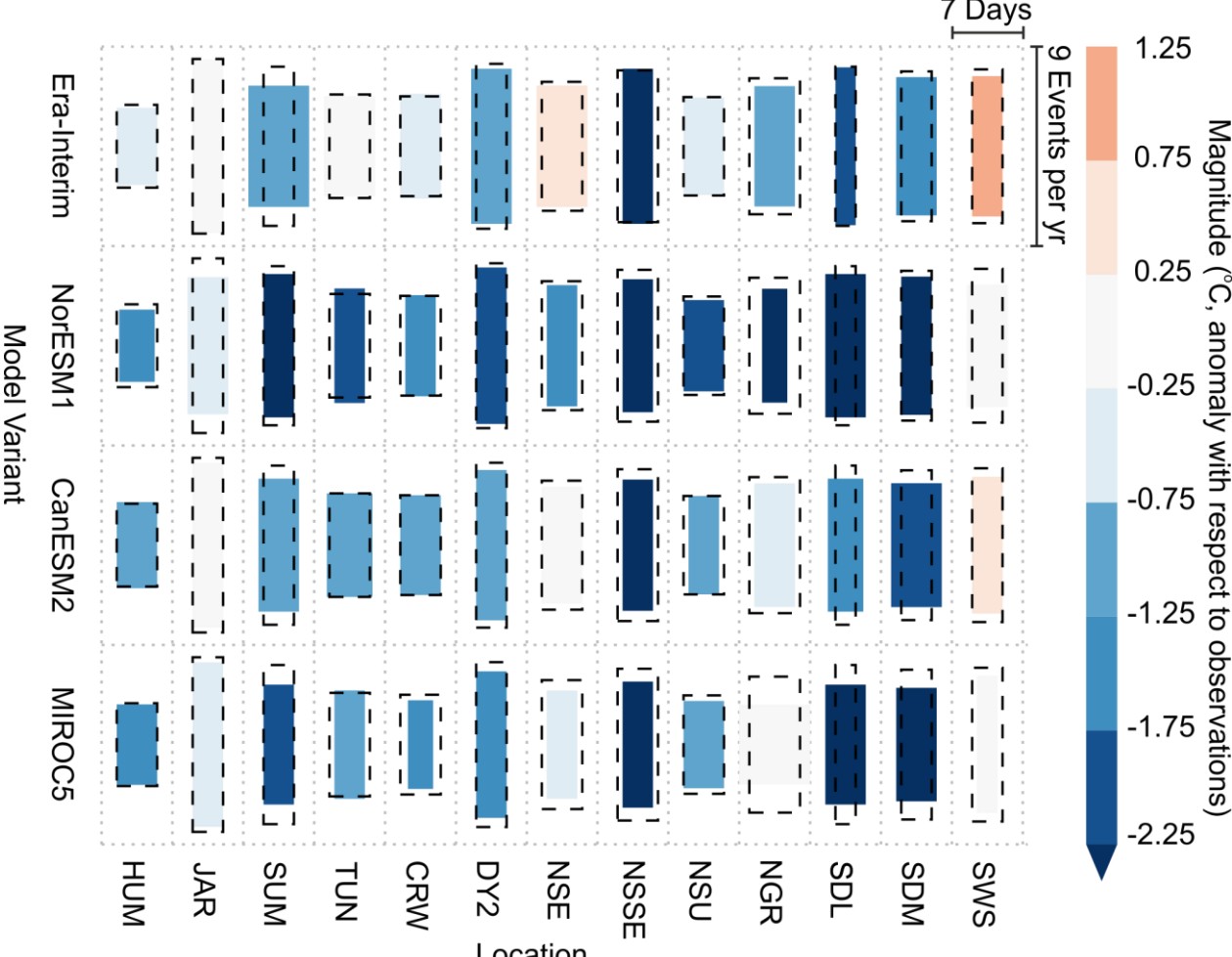

**Figure 3. Median frequency, duration and magnitude of extreme temperature events as simulated by each model variant at each station. Frequency is denoted by the height of each box, duration is indicated by the width of each box, and observed values are given by the dashed black boxes. Box colours indicate the departure of the modelled magnitude from the observed value, blue colours indicate an underestimate, and red colours indicate an over estimate. Note that a temperature lapse rate has been applied to modelled temperatures to account for the difference in elevation between AWS and MAR elevation.**

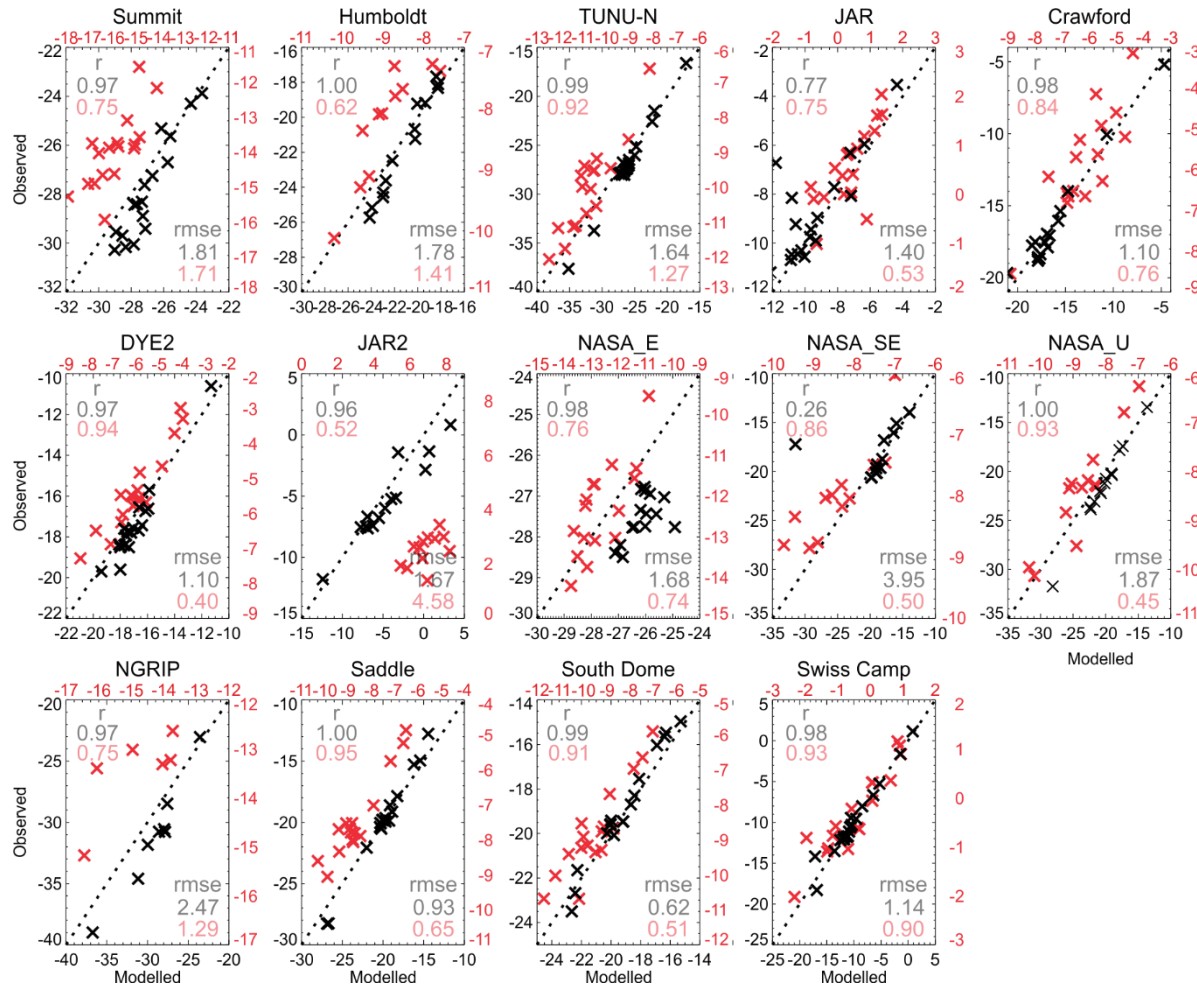

**Figure 4: Modelled vs observed mean annual and mean summer temperature at each of the GC-Net stations. Modelled values are as simulated by MAR-Era. Pearson's correlation co-efficient and the root mean squared error (ºC) between the data is annotated. Red symbols and text refer to summer (JJA) values, black symbols and text refer to annual values. Black dotted line denotes a 1:1 fit.**


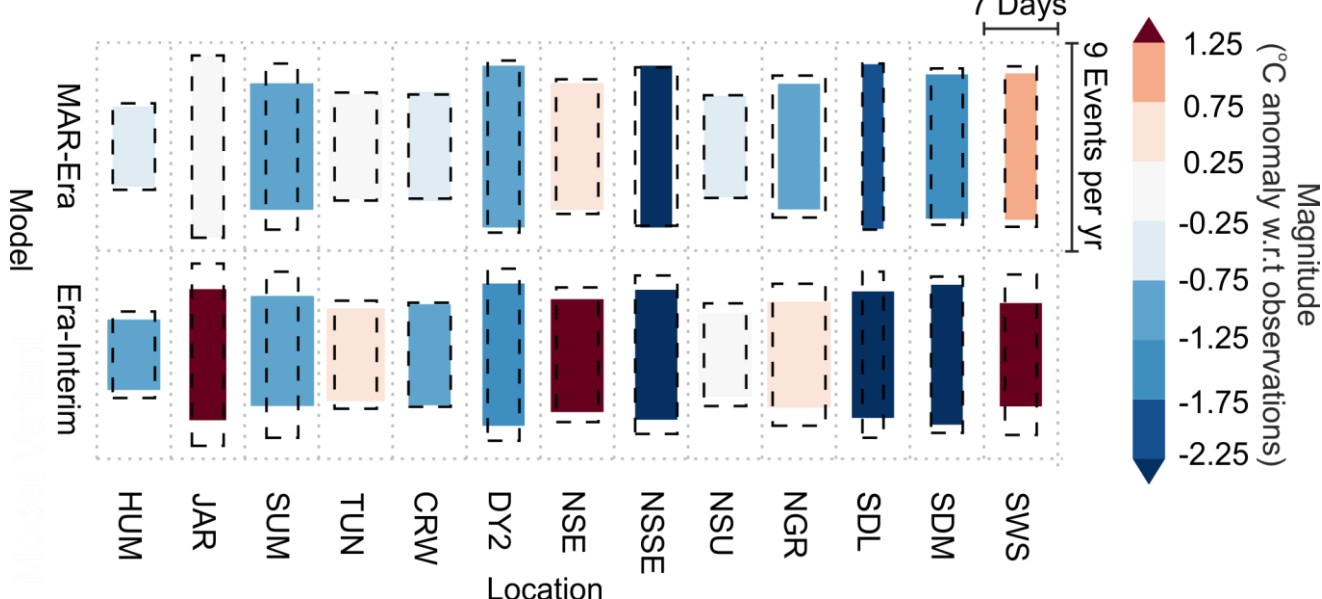

**Figure 5. Median frequency, duration and magnitude of extreme temperature events as simulated by Era-Interim and MAR-Era. As before, frequency is denoted by the height of each box, duration is indicated by the width of each box and observed values are given by the dashed black boxes. Box colours indicate the departure of the modelled magnitude from the observed value, blue colours indicate an underestimate, and red colours indicate an over estimate. Note that a temperature lapse rate has been applied to modelled temperatures to account for differences in elevation between AWS and Era-Interim elevation and AWS and MAR elevation.**

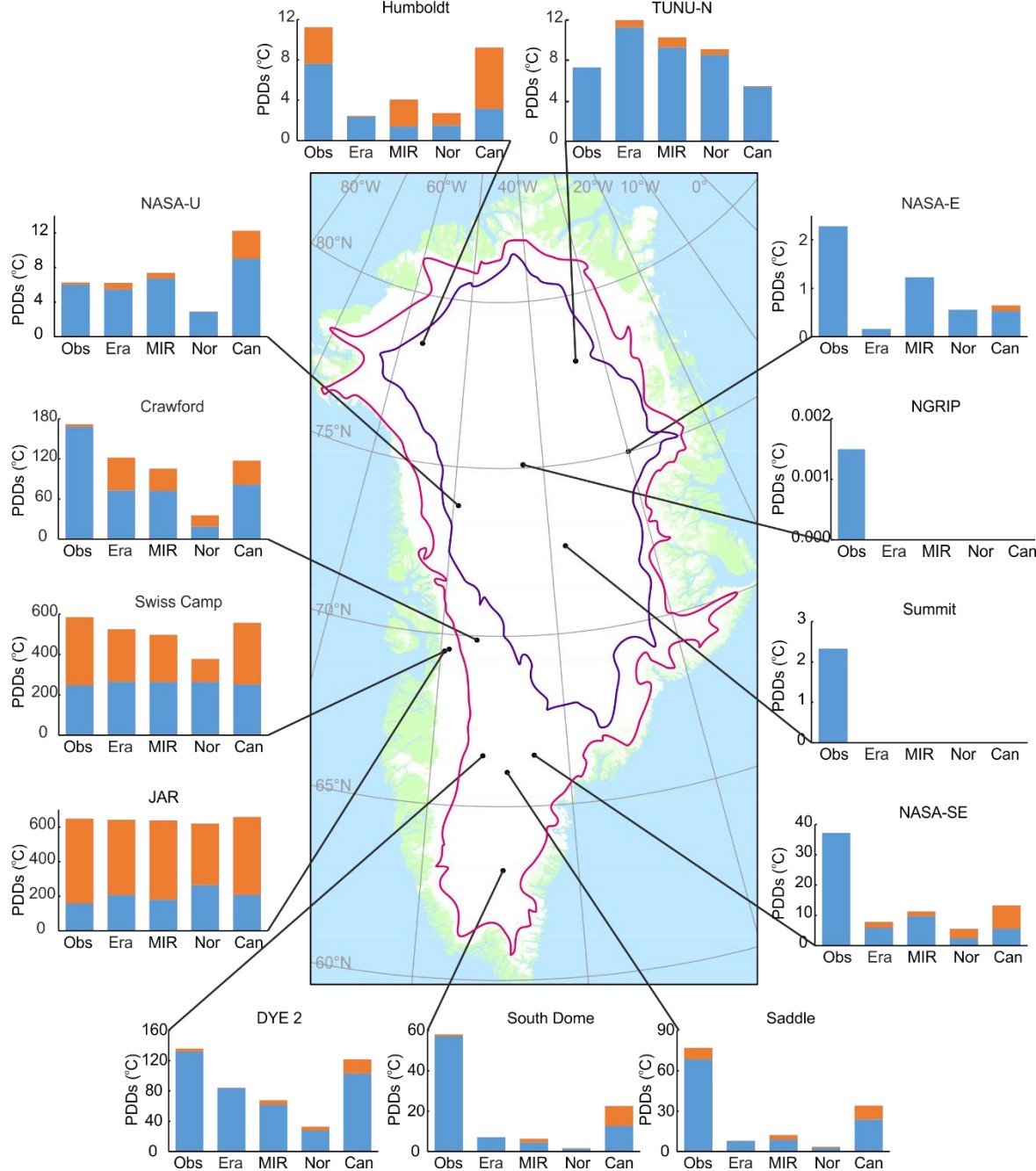

**Figure 6: Total positive degree days observed and modelled at each location during the study period. Stacked bars represent total values. Blue (orange) coloured portion represents contribution during extreme (non-extreme) conditions. Observed values are labelled Obs, model variants are labelled as Era – MAR-Era, MIR – MAR-MIR, Nor – Mar-Nor and Can – MAR-Can. Contours on map represent lower boundary of percolation zone (pink) and dry snow zone (purple). Zones are defined as in McMillan et al., (2016).**

|  | Melt Zone | N years of data | MAR m | Gc-net m | MAR-Gc-net m | Lapse rate corr ºC |
|---|---|---|---|---|---|---|
| Summit | Dry Snow | 16.62 | 3201 | 3254 | -53 | -0.38 |
| JAR | Ablation | 15.93 | 932 | 962 | -30 | -0.21 |
| Humboldt | Dry Snow | 12.44 | 2020 | 1995 | 25 | 0.18 |
| TUNU-N | Dry Snow | 17.38 | 2028 | 2113 | -85 | -0.60 |
| Swiss Camp | Ablation | 15.47 | 1245 | 1149 | 96 | 0.68 |
| Crawford | Percolation | 13.16 | 1920 | 2022 | -102 | -0.72 |
| NASA-U | Dry Snow | 12.48 | 2293 | 2369 | -76 | -0.54 |
| DYE2 | Percolation | 19.58 | 2097 | 2165 | -68 | -0.48 |
| Saddle | Percolation | 17.70 | 2475 | 2559 | -84 | -0.60 |
| South Dome | Percolation | 15.89 | 2833 | 2922 | -89 | -0.63 |
| NASA-E | Dry Snow | 18.13 | 2606 | 2631 | -25 | -0.18 |
| NGRIP | Dry Snow | 7.40 | 2921 | 2950 | -29 | -0.21 |
| NASA-SE | Percolation | 11.55 | 2331 | 2425 | -94 | -0.67 |
| JAR2 | Ablation | 10.85 | 252 | 568 | -316 | -2.25 |

**Table 1: Discrepancy between elevation of AWS and elevation of the closest grid cell in MAR. Number of years of data**
**(excluding gaps) is also given. A lapse rate based correction to MAR temperature data (assuming 0.71ºC per 100 m Steffen and Box (2001)) is also given; negative values occur when the grid cell in MAR is lower down in the atmosphere, i.e. is too warm.**

|            | Observations | MAR-Era | MAR-Can | MAR-MIR | MAR-Nor |
|------------|--------------|---------|---------|---------|---------|
| Summit     | -8.26        | -8.7    | -8.22   | -8.67   | -10.09  |
| JAR        | 2.99         | 3.03    | 3       | 2.83    | 2.43    |
| Humboldt   | -3.24        | -3.56   | -3.74   | -4.18   | -4.93   |
| TUNU-N     | -4.86        | -4.79   | -5.01   | -5.04   | -5.61   |
| Swiss Camp | 2.44         | 1.89    | 1.97    | 1.85    | 1.28    |
| Crawford   | -0.55        | -0.68   | -0.42   | -0.98   | -1.9    |
| NASA-U     | -3.3         | -2.81   | -3.13   | -3.45   | -4.11   |
| DYE2       | -0.33        | -0.81   | -0.79   | -1.24   | -1.76   |
| Saddle     | -2.24        | -2.98   | -3.27   | -3.64   | -4.22   |
| South Dome | -3.12        | -3.63   | -3.84   | -4.73   | -4.91   |
| NASA-E     | -7.39        | -7      | -6.85   | -6.88   | -8.41   |
| NGRIP      | -8.74        | -8.44   | -7.97   | -8.64   | -9.76   |
| NASA-SE    | -2.15        | -2.62   | -3.01   | -2.94   | -3.8    |
| JAR2       | 4.89         | 13.51   | 12.95   | 12.88   | 11.93   |

**Table 2: Extreme event threshold (90$^{th}$ percentile) in each time series.**

| | Humboldt | Jar | Summit | Tunu | Crawford | DYE2 | JAR2 | NASA-E | NASA-SE | NASA-U | NGRIP | Saddle | South Dome | Swiss Camp |
|---|---|---|---|---|---|---|---|---|---|---|---|---|---|---|
| Number of Events | 71 | 134 | 122 | 84 | 81 | 134 | 106 | 99 | 89 | 71 | 43 | 122 | 115 | 125 |
| Frequency (#/yr) | 3.74 | 7.88 | 7.18 | 4.67 | 4.5 | 7.44 | 8.15 | 5.82 | 6.85 | 4.44 | 6.14 | 7.18 | 6.76 | 6.94 |
| Maxima (ºC) | | | | | | | | | | | | | | |
| Minimum | -3.23 | 3.23 | -8.16 | -4.75 | -0.43 | -0.3 | 4.9 | -7.38 | -2.14 | -3.1 | -8.73 | -2.23 | -3.08 | 2.45 |
| 25% quantile | -1.64 | 3.55 | -7.04 | -3.24 | 0.13 | 0.53 | 5.37 | -6.22 | -0.86 | -2.08 | -7.05 | -1.27 | -2.04 | 2.94 |
| Median | -0.64 | 4.1 | -5.07 | -2.14 | 0.95 | 1.32 | 5.94 | -4.19 | 0.55 | -1.04 | -5.7 | 0.16 | -0.36 | 3.43 |
| 75% quantile | 0.02 | 4.85 | -2.67 | -0.41 | 2.02 | 1.97 | 6.63 | -2.3 | 2.14 | 0.18 | -3.87 | 1.41 | 0.88 | 4.36 |
| Maximum | 3.07 | 9.89 | 3.62 | 2.9 | 12.68 | 5.16 | 9.59 | 3.63 | 5.52 | 2.05 | 0.04 | 9.18 | 10.84 | 7.77 |
| Duration (days) | | | | | | | | | | | | | | |
| Minimum | 1 | 1 | 1 | 1 | 1 | 1 | 1 | 1 | 1 | 1 | 1 | 1 | 1 | 1 |
| 25% quantile | 2 | 1 | 1 | 2 | 1 | 1 | 1 | 1 | 2 | 2 | 2 | 1 | 1 | 1 |
| Median | 4 | 3 | 3 | 4 | 4 | 3 | 3 | 4 | 4 | 4 | 5 | 2 | 3 | 3 |
| 75% quantile | 8 | 5 | 10 | 9 | 7 | 7 | 5 | 9 | 8 | 8 | 8 | 8 | 8 | 6 |
| Maximum | 42 | 46 | 31 | 54 | 67 | 36 | 22 | 59 | 29 | 35 | 29 | 39 | 37 | 34 |


**Table 3: Frequency, duration and magnitude of extreme events observed at each of the 14 stations.**

| | ALL nyrs = 233 | | Ablation nyrs = 40 | | Percolation nyrs = 91 | | Dry Snow nyrs=102 | |
|---|---|---|---|---|---|---|---|---|
| All data | $\bar{T}$ | $\dfrac{dT}{dt}$ | $\bar{T}$ | $\dfrac{dT}{dt}$ | $\bar{T}$ | $\dfrac{dT}{dt}$ | $\bar{T}$ | $\dfrac{dT}{dt}$ |
| MAR-Can | -0.13 | 0.03 | -1.17 | -0.05 | -0.06 | -0.01 | 0.22 | 0.11 |
| MAR-MIR | 1.19 | 0.02 | -0.24 | -0.08 | 1.11 | -0.02 | 1.81 | 0.09 |
| MAR-Nor | -0.21 | 0.03 | -1.41 | -0.02 | -0.46 | -0.01 | 0.49 | 0.09 |
| MAR-Era | 0.86 | -0.03 | -0.34 | 0.03 | 0.66 | -0.06 | 1.50 | -0.02 |
| JJA only | | | | | | | | |
| MAR-Can | -0.01 | 0.01 | -0.02 | -0.05 | 0.27 | -0.01 | -0.26 | 0.04 |
| MAR-MIR | -0.64 | -0.04 | -0.56 | -0.07 | -0.62 | -0.07 | -0.68 | 0.00 |
| MAR-Nor | -1.58 | 0.04 | -1.55 | 0.00 | -1.57 | 0.06 | -1.60 | 0.04 |
| MAR-Era | -0.35 | 0.01 | -0.42 | 0.02 | -0.03 | 0.01 | -0.62 | 0.01 |

**Table 4: Modelled-observed values for mean (ºC) and rate of change (ºCyr⁻¹) of mean daily temperature during our study period. Values given in red (blue) denote an over (under) estimate of temperature in general; i.e. the model is too warm (cold). The number of years of data in total (i.e. the sum of the number of years of data at each station) is also identified (nyrs).**

| | | All data | | | | | Extreme Events | | | | | Non Extremes | | | | |
|---|---|---|---|---|---|---|---|---|---|---|---|---|---|---|---|---|
| | | Observed | Era-Interim | MIROC5 | Nor-ESM1 | Can-ESM2 | Observed | Era-Interim | MIROC5 | Nor-ESM1 | Can-ESM2 | Observed | Era-Interim | MIROC5 | Nor-ESM1 | Can-ESM2 |
| Dry snow zone (0.8mil Km$^2$) | PDDs (ºC) | 4.91 | 3.50 | 3.82 | 2.40 | 4.34 | 4.26 | 3.21 | 3.11 | 2.25 | 3.01 | 0.65 | 0.29 | 0.71 | 0.16 | 1.33 |
| | Bias (%) | | -29 | -22 | -51 | -11 | | -25 | -27 | -47 | -29 | | -55 | 10 | -76 | 107 |
| Percolation zone (0.61mil km$^2$) | PDDs (ºC) | 96.04 | 45.84 | 40.41 | 15.10 | 57.74 | 92.55 | 35.53 | 31.27 | 10.51 | 45.20 | 3.49 | 10.31 | 9.14 | 4.60 | 12.54 |
| | Bias (%) | | -52 | -58 | -84 | -40 | | -62 | -66 | -89 | -51 | | 195 | 162 | 32 | 259 |
| Ablation zone (0.23mil km$^2$) | PDDs (ºC) | 616.52 | 583.38 | 544.84 | 479.68 | 573.57 | 202.65 | 236.42 | 221.85 | 263.34 | 229.68 | 413.87 | 346.95 | 322.98 | 216.34 | 343.89 |
| | Bias (%) | | -5 | -12 | -22 | -7 | | 17 | 9 | 30 | 13 | | -16 | -22 | -48 | -17 |
| All Greenland (weighted) | PDDs (ºC) | 124.41 | 100.46 | 93.20 | 73.99 | 103.91 | 64.80 | 47.88 | 44.21 | 41.90 | 50.42 | 59.60 | 52.58 | 48.99 | 32.10 | 53.49 |
| | Bias (%) | | -19 | -25 | -41 | -16 | | -26 | -32 | -35 | -22 | | -12 | -18 | -46 | -10 |

**Table 5: Total positive degree days during the study period. Per station averages are given for the dry snow (six stations), percolation (five stations) and ablation (two stations) zones. These values are weighted by relative area to give a value for the whole of Greenland. The area of each zone is given in each row label, and is in units of million square kilometres.**

