# Peer review of "Extreme temperature events on Greenland in observations and the MAR regional climate model"

_The Cryosphere, 2017_

## Referee Comment (RC2) · Anonymous Referee #2 · 8 Sep 2017

This paper investigates the ability of MAR, a widely used regional climate and Greenland ice sheet surface mass balance model, to simulate extreme near surface temperatures. To this end, MAR outputs are compared with data from a network of automatic weather stations known as GC-Net. It is found that MAR can capture the frequency and duration of extreme temperature events, but underestimates the absolute values of extreme temperature events. Furthermore, the impacts of the underestimated extreme events on Greenland ice sheet melt energy is assessed and translate to between 16-40% underestimation of melt energy.

This paper presents a fascinating and important topic. The idea to compare GC-net stations with MAR simulations to examine extreme temperatures is great. The text is easy to read, but some figures are difficult to interpret and are missing basic information

such as units and scales. There are some additional issues with the analysis and methods in the manuscript as outlined below. With some more work this paper could be an important contribution to better understanding the Greenland ice sheet in a changing climate.

===========

Major comments

===========

1. Using the GC net stations as representative samples of ablation, percolation and dry snow zone is problematic. Rather than being representative of a zone, they can be considered representative for a geographic region of varying sizes. The three stations from the ablation zone are in close proximity of each other on the west coast. All percolation zone stations are in the south, all dry snow zone stations are in the north. Some more rigorous analysis is needed before these stations can be assumed representative of the three regions if at all. I suggest another approach that focuses on model and station comparison rather that generalizing over the three zone. If the authors want to generalize about the three zone a more rigorous analysis of the representability of the stations for each of the zones are needed.

2. Provide information about time span in addition to time series length for each of the GCnet stations and discuss implications of varying time series length and time period span on the extreme value statistics.

3. The analysis of melt energy and extreme temperature events needs some work because extreme temperatures at several of the stations appear to not be associated with melting at all (i.e. Figure 1).

4. A clear presentation of the analysis behind the conclusions that MAR simulates duration of extreme temperature events but not frequency or magnitude of those events are not well supported with figures and tables. It seems that Figure 2 and 4 are indented for

this purpose, but they are not clear (see more comments on the figure design below).

5. The analysis discussed in lines L242 to L248 belongs in the result section and needs some more elaboration to be convincing. First, a figure showing the amount of melt energy during extreme versus average conditions would be really nice to see. Second, you have to address the fact that some stations have extreme temperatures that are not above freezing and therefore no PDDs.

===========

Minor comments

===========

L14: Clarify that you are examining extreme positive temperature anomalies (as opposed to positive and negative)

L92: Clarify what MAR grid cell elevation that lower than the AWS, e.g. the center point?

L105: Explain PDD, the concept may not be widely known outside glaciologist circles.

L108: Provide more background for equation 1. Typically PDD are a function of mean daily temperatures and a temperature to melt conversion factor.

L135: Please write more about Figure 2. It is difficult to interest and draw conclusions from it beyond that MAR data are colder than GC-net station data.

L140-149: Is this data shown in figures or tables? If so clarify how, if not considering adding a figure to clearly illustrate these findings.

L154: Remove the gaps from the total count of data points to give an accurate count of the number of years with data (in table 1).

L170: Explain "raw Era-interim" and how it is different form "Era-interim"

L180: Provide units for PDD here and elsewhere
Figure 1: Show the spatial distribution of the ablation, percolation and accumulation zones

Figure 2: This figure needs work. First, a scale is needed to be able to interpret the height and width of the boxes. Second, it is unclear why the bars are centered in each box. Third, units on the colorbar are needed. Forth, five shades of blue are displayed in the figure, extend the colorbar to capture the darkest blue. Finally, the figure captions says that the figure shows model results, but the colorbar text says it is showing deviations from observations.

Figure 3. Clarify the meaning of the black dotted line. Add units to each axis. It would be useful to also show the RMSE in the plots.

Figure 4. Same comments as Figure 3.

Figure 5: Add units to x-axes

Table 3: Explain the "nyrs" variable. Clarify what spatial domain the data values are calculated for.

Table 4: Provide units for PDD.

---

## Author Comment (AC1)

**Detailed response to Anonymous Referee #2**

Note: reviewer comments are given in bold and our response is given in normal type.

**1. Using the GC net stations as representative samples of ablation, percolation and dry snow zone is problematic. Rather than being representative of a zone, they can be considered representative for a geographic region of varying sizes. The three stations from the ablation zone are in close proximity of each other on the west coast. All percolation zone stations are in the south, all dry snow zone stations are in the north. Some more rigorous analysis is needed before these stations can be assumed representative of the three regions if at all. I suggest another approach that focuses on model and station comparison rather that generalizing over the three zone. If the authors want to generalize about the three zone a more rigorous analysis of the representability of the stations for each of the zones are needed.**

We agree with the reviewer that our station groupings are geographically proximal, however these groupings do also coincide with the ice sheet's melt zones; particularly the dry snow zone, which is mostly in the North of the ice sheet, and the percolation zone, the largest expanse of which is in the South. We have added contours to Figures 1 and 5 to show the coverage of these zones with respect to the distribution of stations.

We appreciate that there are only two stations in the ablation zone, albeit located in the region where a majority of ice ablation occurs (McMillan et al., 2016). We acknowledge the sparsity of the Gc-Net stations with respect to the size of the ice sheet on line 204 of the original manuscript and have added the following in line 220:

'Because we only have data for two ablation zone stations which are located in close proximity, further work is required to assess whether this is a general property of the ablation zone or restricted to this location; temperatures in general are much warmer here…'

**2. Provide information about time span in addition to time series length for each of the GCnet stations and discuss implications of varying time series length and time period span on the extreme value statistics.**

We have included a new figure (the new figure 1) to show the data coverage at the GC-Net stations used here, and amended Table 1 such that it shows the total amount of data (excluding gaps). We have also added text to introduce the new figure, refer explicitly to the table and discuss the significance of the missing data beginning on line 70 as follows:

'Our analysis focuses on 14 of the 18 stations; we found the remaining 4 stations to have temperature time series which were either too short or too patchy for robust statistical analysis. Figure 1 shows the data coverage at each of the 14 stations studied here and Table 1 gives the total number of years of data available, when gaps are excluded. We attribute these missing data to equipment failure and assume that it is unrelated to the occurrence of extreme high temperatures. As such we treat these data gaps as `missing at random' and ignore them in our analysis. Since most of the missing periods cover whole years, rather than just a summer- or winter-period, this assumption is reasonable. '

It is true that varying time series length and span may have implications when comparing extreme event characteristics between sites; for example it is difficult to pick out a trend in extremes in shorter time series. In this study we did not find any evidence of a trend in extremes at any of the stations and in any case the main focus of our paper is on the comparison between GC-Net and MAR, both of which were commonly sampled at each station and subject to the same EVA procedure. We are hoping to investigate temporal trends in extremes in future work however and so we have amended line 260 to read:

'further work is needed to determine the relative contributions of potential physical drivers of extreme events at different locations and over different time periods'

**3. The analysis of melt energy and extreme temperature events needs some work because extreme temperatures at several of the stations appear to not be associated with melting at all (i.e. Figure 1).**

We are not sure what the reviewer is asking for here as Figure 1 does not consider melting; it presents extreme event characteristics at each of the stations.

Figure 5 shows PDDs at each station and attributes them to extreme events (blue portion of stacked bar) or normal conditions (orange portion of stacked bar). Extreme events produce PDDs at all of the GC-Net stations we consider in this study. The MAR model variants do not produce any melting at some of these stations. This is discussed in the manuscript on lines 199-201 and 250-256.

**4. A clear presentation of the analysis behind the conclusions that MAR simulates duration of extreme temperature events but not frequency or magnitude of those events are not well supported with figures and tables. It seems that Figure 2 and 4 are indented for this purpose, but they are not clear (see more comments on the figure design below).**

We have edited the figures in line with the reviewer's suggestions (see specific comments below) and added an additional reference to Figure 2 at line 143. This should be clearer now.

**5. The analysis discussed in lines L242 to L248 belongs in the result section and needs some more elaboration to be convincing. First, a figure showing the amount of melt energy during extreme versus average conditions would be really nice to see. Second, you have to address the fact that some stations have extreme temperatures that are not above freezing and therefore no PDDs.**

These points are discussed on lines 186-201 in the results section and the corresponding data is presented in Table 4. The stacked bars in Figure 5 also illustrate the PDDs during extreme (in blue) vs non-extreme (in orange) conditions. Hopefully the edits we have made to this figure will make this point clearer.

**Minor comments**
**L14: Clarify that you are examining extreme positive temperature anomalies (as opposed to positive and negative)**

Edited to read 'extreme positive temperature'

**L92: Clarify what MAR grid cell elevation that lower than the AWS, e.g. the center point?**

'data' changed to 'cell centre' to clarify.

**L105: Explain PDD, the concept may not be widely known outside glaciologist circles.**

Edited to read: 'Melting is most appropriately calculated as a function of the surface energy balance; however measurements of variables required to calculate the surface energy balance (e.g. net radiation, wind speed) are not consistently available at the Gc-Net stations. Positive Degree Days are an estimate of the magnitude and duration of above-zero temperature events and are typically well-correlated with melting (e.g. Braithwaite (1995),

Huybrechts et al. (1991)). Here we calculate positive degree-days (PDD) for both observed and modelled temperatures and take this to be a reasonable approximation for melt energy'

**L108: Provide more background for equation 1. Typically PDD are a function of mean daily temperatures and a temperature to melt conversion factor.'**

We choose to calculate PDDs by integrating a model of daily temperature variability because we are interested in potential melt that occurs during unusually warm conditions. It may be that the mean temperature for a day is 'normal' but the daytime has been unusually warm and the night time has been unusually cold, for example during persistent high pressure conditions in summer. As such we could 'miss' melting that occurs during the warmer part of the day if we use an equation based on mean daily temperatures. This concept is not new as a similar approach has been applied to higher resolution (i.e. sub-daily) data in studies of Antarctica (e.g. Vaughan, 2006, Barrand et al., 2013)

**L135: Please write more about Figure 2. It is difficult to interest and draw conclusions from it beyond that MAR data are colder than GC-net station data.**

Figure 2 is also discussed in the remainder of this paragraph, we have added an additional reference to the figure on line 143 to clarify.

**L140-149: Is this data shown in figures or tables? If so clarify how, if not considering adding a figure to clearly illustrate these findings.**

The text on lines 140-149 refers to Figure 2. A citation to the figure has been added on line 143, and more information has been added to the figure to clarify.

**L154: Remove the gaps from the total count of data points to give an accurate count of the number of years with data (in table 1).**

Done.

**L170: Explain "raw Era-interim" and how it is different form "Era-interim"**

We use the term raw Era-Interim in order to clearly distinguish from MAR forced by Era-Interim. Have edited to clarify as follows: '…raw Era-Interim output (i.e. not MAR forced with Era-Interim) captures…'.

**L180: Provide units for PDD here and elsewhere**

PDDs are in $^{o}$C, edited throughout to clarify

**Figure 1: Show the spatial distribution of the ablation, percolation and accumulation zones**

This information has been added to Figure 1 and Figure 5.

**Figure 2: This figure needs work. First, a scale is needed to be able to interpret the height and width of the boxes.**

Scale added.

**Second, it is unclear why the bars are centered in each box.**

The bars are centred in each box the better to contrast the difference with the observations.

**Third, units on the colorbar are needed.**

Units added

**Forth, five shades of blue are displayed in the figure, extend the colorbar to capture the darkest blue.**

The darkest blue is the bottom out of bounds colour. Apologies; this fell off the submitted version.

**Finally, the figure captions says that the figure shows model results, but the colorbar text says it is showing deviations from observations.**

Yes, it is the modelled magnitude – the observed magnitude, i.e. the model anomaly with respect to the observations. Caption text edited to be: 'of the modelled magnitude' to clarify.

**Figure 3. Clarify the meaning of the black dotted line. Add units to each axis. It would be useful to also show the RMSE in the plots.**

Text added to figure caption: 'Black dotted line denotes a 1:1 fit.'
RMSE added as annotation. Caption edited to read: 'Pearson's correlation co-efficient and the root mean squared error (°C) between the data is annotated'

**Figure 4. Same comments as Figure 3.**

We assume that you mean the same as Figure 2 and have edited accordingly.

**Figure 5: Add units to x-axes**

Edited as requested

**Table 3: Explain the "nyrs" variable. Clarify what spatial domain the data values are calculated for.**

Text added to figure caption: 'The number of years of data in total (i.e. the sum of the number of years of data at each station) is also identified (nyrs).'

The spatial domain for each melt zone is taken to be that defined in McMillan et al., 2016, this is specified in lines 112-115 in the 'methods' section.

**Table 4: Provide units for PDD**

Edited as requested

**References**

Barrand, N. E., Vaughan, D. G., Steiner, N., Tedesco, M., Munneke, P. K., Van Den Broeke, M. R. & Hosking, J. S. 2013. Trends in Antarctic Peninsula surface melting conditions from observations and regional climate modeling. *Journal of Geophysical Research-Earth Surface,* 118**,** 315-330.

Vaughan, D. G. 2006. Recent trends in melting conditions on the Antarctic Peninsula and their implications for ice-sheet mass balance and sea level. *Arctic Antarctic and Alpine Research,* 38**,** 147-152.

---

## Author Comment (AC2)

**Detailed response to Anonymous Referee #2**

Note: reviewer comments are given in bold and our response is given in normal type.

**page 1, line 12 (Abstract) "short period variability in time" - please quantify what time range you mean here.**

Text added: '(i.e. intra-seasonal)'

**p.1, l.24 Re. GrIS recent significant mass loss, please add the following two highly relevant, more recent references to Shepherd et al. 2012:**

*Hanna, Edward and Navarro, Francisco J. and Pattyn, Frank and Domingues, Catia M. and Fettweis, Xavier and Ivins, Erik R. and Nicholls, Robert J. and Ritz, Catherine and Smith, Ben and Tulaczyk, Slawek and Whitehouse, Pippa L. and Jay Zwally, H. (2013) Ice-sheet mass balance and climate change. Nature, 498 (7452). pp. 51-59.*

*van den Broeke, M. R., Enderlin, E. M., Howat, I. M., Kuipers Munneke, P., Noël, B. P. Y., van de Berg, W. J., van Meijgaard, E., and Wouters, B.: On the recent contribution of the Greenland ice sheet to sea level change, The Cryosphere, 10, 1933-1946, https://doi.org/10.5194/tc-10-1933-2016, 2016.*

Both citations added alongside Shepherd et al., 2012 and Hanna et al., added into reference list.

**p.1, l.27 Re. recent episodes of rare and extreme surface melt (2012), please add the following two highly relevant references to Ngheim et al. (2012):**

*Tedesco, M., Fettweis, X., Mote, T., Wahr, J., Alexander, P., Box, J. E., and Wouters, B.: Evidence and analysis of 2012 Greenland records from spaceborne observations, a regional climate model and reanalysis data, The Cryosphere, 7, 615-630, https://doi.org/10.5194/tc-7-615-2013, 2013.*

*Hanna, Edward and Fettweis, X. and Mernild, S. H. and Cappelen, J. and Ribergaard, M. H. and Shuman, C. A. and Steffen, K. and Wood, L. and Mote, T. L. (2014) Atmospheric and oceanic climate forcing of the exceptional Greenland ice sheet surface melt in summer 2012. International Journal of Climatology, 34 (4). pp. 1022-1037.*

Both citations added alongside Nghiem et al., and added into reference list.

**page 2, line 39: after Noel et al., 2016 reference, suggest inert new sentence: "Alternative statistical downscaling techniques fulfill a similar purpose and give broadly comparable results (Wilton et al. 2017, Vernon et al. 2013), and add the following two relevant references. However, RCMs can also make...":**

*Wilton, D. J. and Jowett, A. and Hanna, E. and Bigg, G. R. and Van Den Broeke, M. R. and Fettweis, X. and Huybrechts, P. (2017) High resolution (1 km) positive degree-day modelling of Greenland ice sheet surface mass balance, 1870-2012 using reanalysis data. Journal of Glaciology, 63 (237). pp. 176-193.*

*Vernon, C. L., Bamber, J. L., Box, J. E., van den Broeke, M. R., Fettweis, X., Hanna, E., and Huybrechts, P.: Surface mass balance model intercomparison for the Greenland ice sheet, The Cryosphere, 7, 599-614, https://doi.org/10.5194/tc-7-599-2013, 2013.*

Text edited as suggested including citations. Both papers added to reference list.

**p.2, l.45 change to "fidelity at the regional OR SEASONAL scales does not...".**

Edited as requested

**p.2, l.53: GC-Net also needs Steffen et al. reference here.**

Citation added.

**p.3, l.69 change "max" to "maximum".**

Edited as requested

**p.3, ll.74/75 slightly reword to "The MAR version 3.5 used here has been EXTENSIVELY evaluated in...".**

Edited as requested

**p.3, l.87 "MAR-Era data ARE available...".**

Edited as requested

**p.4, ll.97 & 99: "data set" -> "dataset".**

Edited as requested.

**p.4, l.101 "below the threshold for three consecutive days"- based on daily mean temperature (and, if so, how is the latter calculated?) or what exactly? Needs a bit more detail/explanation since how this is defined can affect the results.**

We use the maximum daily temperature time series to identify extreme events. We choose to use the maximum, rather than the mean, temperatures in order to capture high temperature events that may last for < 1 day.

Text edited on line 99 to read: 'type-specific threshold applied to the maximum daily temperature time series' and on line 100 to read: 'start once the maximum daily temperature'

**p.5, l.127: Why is event frequency \*positively\* correlated with elevation in North Greenland/the dry snow zone?**

We have yet to identify a satisfactory explanation for this which is why we decline to comment in the manuscript. We wonder if perhaps it is to do with increased exposure with elevation but further investigation is required to ascertain whether this is in fact the case.

**p.5, l.154: change "~decadal scale" to "decadal scale".**

Edited as requested

**p.6, l.158: change ">1.5°C" to ">=1.5°C" since MAR_Era = 1.50°C.**

Edited as requested.

**p.7, ll.216-218: "extreme melt years on Greenland have been attributed to an increase in the frequency and duration of high pressure conditions...Greenland Blocking Index" -please add the two highly relevant citations and add them in the reference list:**

*Hanna, Edward and Cropper, Thomas E. and Hall, Richard J. and Cappelen, John (2016) Greenland Blocking Index 1851-2015: a regional climate change signal. International Journal of Climatology, 36 (15). pp. 4847-4861.*

*Hanna, E. and Jones, J. M. and Cappelen, J. and Mernild, S. H. and Wood, L. and Steffen, K. and Huybrechts, P. (2013) The influence of North Atlantic atmospheric and oceanic forcing effects on 1900-2010 Greenland summer climate and ice melt/runoff. International Journal of Climatology, 33 (4). pp. 862-880.*

Citations added as requested, both in text and to reference list.

**p.8, l.247 needs punctuation correction to "...given its assimilation of observations; however, we note that...".**

Edited as requested.

**p.9, l.259: "strongly controlled by geography" sounds a bit vague. Can you be more specific, e.g. say topography, elevation and ice/snow facies etc.?**

ok, edited to read: '…geography (e.g. topography, elevation, latitude etc), though…'

**p.18/Table 3: add units (e.g. ºC/yr?) for "rate of change of mean daily temperature".**

Edited as requested.

---

## Author Response (ED1)

**Author's Response to Peer Review**

On behalf of all of the authors I would like to thank both reviewers for their detailed and insightful comments. We feel that their advice has improved the manuscript and in particular we have:

- Improved our attribution of relevant background literature
- Edited our figures and tables so that their contents are easier to understand
- Included a new figure detailing the data coverage at each station (new figure 1)

Our response to specific comments follows, note that the reviewer comment is given in bold and our response is given in normal type. References to page numbers refer to the original manuscript.

Best wishes

Amber Leeson
* * *
**Reviewer #1**

**page 1, line 12 (Abstract) "short period variability in time" - please quantify what time range you mean here.**

Text added: '(i.e. intra-seasonal)'

**p.1, l.24 Re. GrIS recent significant mass loss, please add the following two highly relevant, more recent references to Shepherd et al. 2012:**

*Hanna, Edward and Navarro, Francisco J. and Pattyn, Frank and Domingues, Catia M. and Fettweis, Xavier and Ivins, Erik R. and Nicholls, Robert J. and Ritz, Catherine and Smith, Ben and Tulaczyk, Slawek and Whitehouse, Pippa L. and Jay Zwally, H. (2013) Ice-sheet mass balance and climate change. Nature, 498 (7452). pp. 51-59.*

*van den Broeke, M. R., Enderlin, E. M., Howat, I. M., Kuipers Munneke, P., Noël, B. P. Y., van de Berg, W. J., van Meijgaard, E., and Wouters, B.: On the recent contribution of the Greenland ice sheet to sea level change, The Cryosphere, 10, 1933-1946, https://doi.org/10.5194/tc-10-1933-2016, 2016.*

Both citations added alongside Shepherd et al., 2012 and Hanna et al., added into reference list.

**p.1, l.27 Re. recent episodes of rare and extreme surface melt (2012), please add the following two highly relevant references to Ngheim et al. (2012):**

*Tedesco, M., Fettweis, X., Mote, T., Wahr, J., Alexander, P., Box, J. E., and Wouters, B.: Evidence and analysis of 2012 Greenland records from spaceborne observations, a regional climate model and reanalysis data, The Cryosphere, 7, 615-630, https://doi.org/10.5194/tc-7-615-2013, 2013.*

*Hanna, Edward and Fettweis, X. and Mernild, S. H. and Cappelen, J. and Ribergaard, M. H. and Shuman, C. A. and Steffen, K. and Wood, L. and Mote, T. L. (2014) Atmospheric and oceanic climate forcing of the exceptional Greenland ice sheet surface melt in summer 2012. International Journal of Climatology, 34 (4). pp. 1022-1037.*

Both citations added alongside Nghiem et al., and added into reference list.

**page 2, line 39: after Noel et al., 2016 reference, suggest inert new sentence: "Alternative statistical downscaling techniques fulfill a similar purpose and give broadly comparable results (Wilton et al. 2017, Vernon et al. 2013), and add the following two relevant references. However, RCMs can also make...":**

*Wilton, D. J. and Jowett, A. and Hanna, E. and Bigg, G. R. and Van Den Broeke, M. R. and Fettweis, X. and Huybrechts, P. (2017) High resolution (1 km) positive degree-day modelling of Greenland ice sheet surface mass balance, 1870-2012 using reanalysis data. Journal of Glaciology, 63 (237). pp. 176-193.*

*Vernon, C. L., Bamber, J. L., Box, J. E., van den Broeke, M. R., Fettweis, X., Hanna, E., and Huybrechts, P.: Surface mass balance model intercomparison for the Greenland ice sheet, The Cryosphere, 7, 599-614, https://doi.org/10.5194/tc-7-599-2013, 2013.*

Text edited as suggested including citations. Both papers added to reference list.

**p.2, l.45 change to "fidelity at the regional OR SEASONAL scales does not...".**

Edited as requested

**p.2, l.53: GC-Net also needs Steffen et al. reference here.**

Citation added.

**p.3, l.69 change "max" to "maximum".**

Edited as requested

**p.3, ll.74/75 slightly reword to "The MAR version 3.5 used here has been EXTENSIVELY evaluated in...".**

Edited as requested

**p.3, l.87 "MAR-Era data ARE available...".**

Edited as requested

**p.4, ll.97 & 99: "data set" -> "dataset".**

Edited as requested.

**p.4, l.101 "below the threshold for three consecutive days"- based on daily mean temperature (and, if so, how is the latter calculated?) or what exactly? Needs a bit more detail/explanation since how this is defined can affect the results.**

We use the maximum daily temperature time series to identify extreme events. We choose to use the maximum, rather than the mean, temperatures in order to capture high temperature events that may last for < 1 day.

Text edited on line 99 to read: 'type-specific threshold applied to the maximum daily temperature time series' and on line 100 to read: 'start once the maximum daily temperature'

**p.5, l.127: Why is event frequency \*positively\* correlated with elevation in North Greenland/the dry snow zone?**

We have yet to identify a satisfactory explanation for this which is why we decline to comment in the manuscript. We wonder if perhaps it is to do with increased exposure with elevation but further investigation is required to ascertain whether this is in fact the case.

**p.5, l.154: change "~decadal scale" to "decadal scale".**

Edited as requested

**p.6, l.158: change ">1.5ºC" to ">=1.5ºC" since MAR_Era = 1.50ºC.**

Edited as requested.

**p.7, ll.216-218: "extreme melt years on Greenland have been attributed to an increase in the frequency and duration of high pressure conditions...Greenland Blocking Index" -please add the two highly relevant citations and add them in the reference list:**

*Hanna, Edward and Cropper, Thomas E. and Hall, Richard J. and Cappelen, John (2016) Greenland Blocking Index 1851-2015: a regional climate change signal. International Journal of Climatology, 36 (15). pp. 4847-4861.*

*Hanna, E. and Jones, J. M. and Cappelen, J. and Mernild, S. H. and Wood, L. and Steffen, K. and Huybrechts, P. (2013) The influence of North Atlantic atmospheric and oceanic forcing effects on 1900-2010 Greenland summer climate and ice melt/runoff. International Journal of Climatology,*
*33 (4). pp. 862-880.*

Citations added as requested, both in text and to reference list.

**p.8, l.247 needs punctuation correction to "...given its assimilation of observations; however, we note that...".**

Edited as requested.

**p.9, l.259: "strongly controlled by geography" sounds a bit vague. Can you be more specific, e.g. say topography, elevation and ice/snow facies etc.?**

ok, edited to read: '…geography (e.g. topography, elevation, latitude etc), though…'

**p.18/Table 3: add units (e.g. ºC/yr?) for "rate of change of mean daily temperature".**

Edited as requested.
* * *
**Reviewer #2**

We would like to thank the reviewer for their assessment and particularly their kind words in support of our work. We are particularly pleased to hear that they found our submission 'fascinating and important'!

Our response to the reviewer comments is given here with the comment given in bold and our response in normal type. Where edits have been made to the paper on the reviewer's

recommendation, page numbers refer to the corresponding position in the **original** manuscript.

**1. Using the GC net stations as representative samples of ablation, percolation and dry snow zone is problematic. Rather than being representative of a zone, they can be considered representative for a geographic region of varying sizes. The three stations from the ablation zone are in close proximity of each other on the west coast. All percolation zone stations are in the south, all dry snow zone stations are in the north. Some more rigorous analysis is needed before these stations can be assumed representative of the three regions if at all. I suggest another approach that focuses on model and station comparison rather that generalizing over the three zone. If the authors want to generalize about the three zone a more rigorous analysis of the representability of the stations for each of the zones are needed.**

We agree with the reviewer that our station groupings are geographically proximal, however these groupings do also coincide with the ice sheet's melt zones; particularly the dry snow zone, which is mostly in the North of the ice sheet, and the percolation zone, the largest expanse of which is in the South. We have added contours to Figures 1 and 5 to show the coverage of these zones with respect to the distribution of stations.

We appreciate that there are only two stations in the ablation zone, albeit located in the region where a majority of ice ablation occurs (McMillan et al., 2016). We acknowledge the sparsity of the Gc-Net stations with respect to the size of the ice sheet on line 204 of the original manuscript and have added the following in line 220:

'Because we only have data for two ablation zone stations which are located in close proximity, further work is required to assess whether this is a general property of the ablation zone or restricted to this location; temperatures in general are much warmer here…'

**2. Provide information about time span in addition to time series length for each of the GCnet stations and discuss implications of varying time series length and time period span on the extreme value statistics.**

We have included a new figure (the new figure 1) to show the data coverage at the GC-Net stations used here, and amended Table 1 such that it shows the total amount of data (excluding gaps). We have also added text to introduce the new figure, refer explicitly to the table and discuss the significance of the missing data beginning on line 70 as follows:

'Our analysis focuses on 14 of the 18 stations; we found the remaining 4 stations to have temperature time series which were either too short or too patchy for robust statistical analysis. Figure 1 shows the data coverage at each of the 14 stations studied here and Table 1 gives the total number of years of data available, when gaps are excluded. We attribute these missing data to equipment failure and assume that it is unrelated to the occurrence of extreme high temperatures. As such we treat these data gaps as 'missing at random' and ignore them in our analysis. Since most of the missing periods cover whole years, rather than just a summer- or winter-period, this assumption is reasonable. '

It is true that varying time series length and span may have implications when comparing extreme event characteristics between sites; for example it is difficult to pick out a trend in extremes in shorter time series. In this study we did not find any evidence of a trend in extremes at any of the stations and in any case the main focus of our paper is on the comparison between GC-Net and MAR, both of which were commonly sampled at each station and subject to the same EVA procedure. We are hoping to investigate temporal trends in extremes in future work however and so we have amended line 260 to read:

'further work is needed to determine the relative contributions of potential physical drivers of extreme events at different locations and over different time periods'

**3. The analysis of melt energy and extreme temperature events needs some work because extreme temperatures at several of the stations appear to not be associated with melting at all (i.e. Figure 1).**

We are not sure what the reviewer is asking for here as Figure 1 does not consider melting; it presents extreme event characteristics at each of the stations.

Figure 5 shows PDDs at each station and attributes them to extreme events (blue portion of stacked bar) or normal conditions (orange portion of stacked bar). Extreme events produce PDDs at all of the GC-Net stations we consider in this study. The MAR model variants do not produce any melting at some of these stations. This is discussed in the manuscript on lines 199-201 and 250-256.

**4. A clear presentation of the analysis behind the conclusions that MAR simulates duration of extreme temperature events but not frequency or magnitude of those events are not well supported with figures and tables. It seems that Figure 2 and 4 are indented for this purpose, but they are not clear (see more comments on the figure design below).**

We have edited the figures in line with the reviewer's suggestions (see specific comments below) and added an additional reference to Figure 2 at line 143. This should be clearer now.

**5. The analysis discussed in lines L242 to L248 belongs in the result section and needs some more elaboration to be convincing. First, a figure showing the amount of melt energy during extreme versus average conditions would be really nice to see. Second, you have to address the fact that some stations have extreme temperatures that are not above freezing and therefore no PDDs.**

These points are discussed on lines 186-201 in the results section and the corresponding data is presented in Table 4. The stacked bars in Figure 5 also illustrate the PDDs during extreme (in blue) vs non-extreme (in orange) conditions. Hopefully the edits we have made to this figure will make this point clearer.

**Minor comments**
**L14: Clarify that you are examining extreme positive temperature anomalies (as opposed to positive and negative)**

Edited to read 'extreme positive temperature'

**L92: Clarify what MAR grid cell elevation that lower than the AWS, e.g. the center point?**

'data' changed to 'cell centre' to clarify.

**L105: Explain PDD, the concept may not be widely known outside glaciologist circles.**

Edited to read: 'Melting is most appropriately calculated as a function of the surface energy balance; however measurements of variables required to calculate the surface energy balance (e.g. net radiation, wind speed) are not consistently available at the Gc-Net stations. Positive Degree Days are an estimate of the magnitude and duration of above-zero temperature events and are typically well-correlated with melting (e.g. Braithwaite (1995),

Huybrechts et al. (1991)). Here we calculate positive degree-days (PDD) for both observed and modelled temperatures and take this to be a reasonable approximation for melt energy'

**L108: Provide more background for equation 1. Typically PDD are a function of mean daily temperatures and a temperature to melt conversion factor.'**

We choose to calculate PDDs by integrating a model of daily temperature variability because we are interested in potential melt that occurs during unusually warm conditions. It may be that the mean temperature for a day is 'normal' but the daytime has been unusually warm and the night time has been unusually cold, for example during persistent high pressure conditions in summer. As such we could 'miss' melting that occurs during the warmer part of the day if we use an equation based on mean daily temperatures. This concept is not new as a similar approach has been applied to higher resolution (i.e. sub-daily) data in studies of Antarctica (e.g. Vaughan, 2006, Barrand et al., 2013)

**L135: Please write more about Figure 2. It is difficult to interest and draw conclusions from it beyond that MAR data are colder than GC-net station data.**

Figure 2 is also discussed in the remainder of this paragraph, we have added an additional reference to the figure on line 143 to clarify.

**L140-149: Is this data shown in figures or tables? If so clarify how, if not considering adding a figure to clearly illustrate these findings.**

The text on lines 140-149 refers to Figure 2. A citation to the figure has been added on line 143, and more information has been added to the figure to clarify.

**L154: Remove the gaps from the total count of data points to give an accurate count of the number of years with data (in table 1).**

Done.

**L170: Explain "raw Era-interim" and how it is different form "Era-interim"**

We use the term raw Era-Interim in order to clearly distinguish from MAR forced by Era-Interim. Have edited to clarify as follows: '…raw Era-Interim output (i.e. not MAR forced with Era-Interim) captures…'.

**L180: Provide units for PDD here and elsewhere**

PDDs are in $^{\circ}$C, edited throughout to clarify

**Figure 1: Show the spatial distribution of the ablation, percolation and accumulation zones**

This information has been added to Figure 1 and Figure 5.

**Figure 2: This figure needs work. First, a scale is needed to be able to interpret the height and width of the boxes.**

Scale added.

**Second, it is unclear why the bars are centered in each box.**

The bars are centred in each box the better to contrast the difference with the observations.

**Third, units on the colorbar are needed.**

Units added

**Forth, five shades of blue are displayed in the figure, extend the colorbar to capture the darkest blue.**

The darkest blue is the bottom out of bounds colour. Apologies; this fell off the submitted version.

**Finally, the figure captions says that the figure shows model results, but the colorbar text says it is showing deviations from observations.**

Yes, it is the modelled magnitude – the observed magnitude, i.e. the model anomaly with respect to the observations. Caption text edited to be: 'of the modelled magnitude' to clarify.

**Figure 3. Clarify the meaning of the black dotted line. Add units to each axis. It would be useful to also show the RMSE in the plots.**

Text added to figure caption: 'Black dotted line denotes a 1:1 fit.'
RMSE added as annotation. Caption edited to read: 'Pearson's correlation co-efficient and the root mean squared error (°C) between the data is annotated'

**Figure 4. Same comments as Figure 3.**

We assume that you mean the same as Figure 2 and have edited accordingly.

**Figure 5: Add units to x-axes**

Edited as requested

**Table 3: Explain the "nyrs" variable. Clarify what spatial domain the data values are calculated for.**

Text added to figure caption: 'The number of years of data in total (i.e. the sum of the number of years of data at each station) is also identified (nyrs).'

The spatial domain for each melt zone is taken to be that defined in McMillan et al., 2016, this is specified in lines 112-115 in the 'methods' section.

**Table 4: Provide units for PDD**

Edited as requested

[revised manuscript text omitted]

---

## Author Response (AR2)

**Response to editorial comments**

Comments from the editor are given in bold type and our response is given in normal font. Text from the manuscript is given in speechmarks and italicised with additions/replacements underlined.

**Line 11: can you please add references or justify this statement ?**

References to Medley et al., 2013 and Leeson et al, 2017 added.

**Line 13: can you ssay not 'fully' investigated ? there are a few studies looking at max temp.**

Edited to read: *'has not been fully assessed'*

**Line 14: spell in the abstract**

Acronym for GC-Net expanded.

**Line 16: Fahrheneit r celsius ?**

*'Celsius'* added

**Line 17: not sure what this means**

'*Corollary*' replaced with '*result*'.

**Line 20: can you briefly say where the forcing data is used in the RCM ?**

Edited to read: *'passed into MAR from boundary forcing data'*

**Line 30: not sure this is accepted in the cryosphere format**

*'Ibid'* replaced by reference to Nghiem et al.,

**Line 103: what is the spatial resolution of MAR ?**

Edited to read: *'model grid cell (25 km by 25 km resolution) closest'*

**Line 105: which DEM is used for this and how the lapse rate is computed ?**

Edited to read: *'typically at a lower elevation than the AWS, according to the MAR DEM and the measured elevation of the AWS, and so we apply a lapse rate based correction to MAR temperature data (0.71$^o$C per 100 m of elevation difference, Steffen and Box (2001))'*

**Line 113: would it be possible to have a table of these thresholds for the different stations or periods ? It would be nice for the community I guess ...**

Added as new table 2

**Line 122: I would encourage you to look at the recent paper by E. Hanna and his student looking at distributed PDD.**

We assume that the editor is referring to Wilton et al., in his comment here. We are familiar with this work, and indeed it is already cited on line 43.

**Line 125: I assume this is the 2-m temperature. This should be better explained.**

Line 129-131 edited to read: *'Where A is daily maximum temperature, B is daily mean temperature and φ is one day. Daily mean and maximum 2 m temperature  are output by MAR; for GC-Net data daily mean and maximum are calculated based on hourly data as detailed above.'*

**Also, as you are using daily-averaged temperatures , I am wondering what is the sensitivity of PDD to this choice**

The figure below shows MAR simulated 3 hourly temperature (black), daily mean temperature (red), daily max temperature (green) of a pixel at 1200m near K-transect during Aug 2017. It is clear from this figure that a sinus based approximation holds extremely well, except sometimes when temperatures are below 0°C, but in this case we do not calculate any PDDs.

[Figure]

**and how the uncertainty on GC-NET data can influence the fitting. This analysis should be present in a study dealing with extremes.**

The uncertainty on the GC-Net data is +/- 0.1°C. Assuming errors occur at random (which is reasonable because this is not a systematic bias), then this would not have an effect on daily mean GC-Net temperatures averaged from hourly data and thus no effect on fitting the sinusoidal relationship.

Even if mean daily temperatures from GC-Net do have an uncertainty of +/- 0.1°C then this is an order of magnitude smaller than the temperature discrepancy we find between GC-Net and MAR during extremes.

**Line 131: why not using those using MAR considering that MAR is actually used here ?**

We use the RACMO generated definitions here as 1) they are independent and 2) have already been published and so our paper is consistent with previous work.

**Line 133: why 5 mm w.e. ?**

This threshold was based on expert advice from the MAR and RACMO groups at the time the McMillan et al., paper was in preparation.

**Line 150: What is the sensitivity of the events ti the errors associated with GC-Net data ?**

Assuming that the errors on the GC-Net data are random then events are insensitive to this uncertainty.

**Also, can you add uncertainty on the trends ?**

Added.

**Line 164: you need to be specific that ERA is not a GCM but re-analysis ...**

We define the differences between ERA and the GCMs on lines 93-94 but have reiterated that Nor-ESM1, Can-ESM2 and MIROC5 are GCMs here for clarity.

**Line 204: I would bot call the outputs MAR variants as it looks it depends on MAR when in reality it depends on rhe forcing ....**

We are struggling to come up with an alternative phrase to use here and so we would prefer to retain the word 'variant' if possible.

**Line 270: how are statistically robust these results ? There is no sensitivity analysis. Measurements dont imply that they are perfect ... please see my comment above ...**

This is an interesting point, particularly as it refers to a binary (melt vs no melt) finding and we do not wish to be too speculative here. However if we make the conservative, but still potentially realistic,

assumption that all GC-Net measurements are 0.1° C too warm, we still calculate above-zero temperatures and therefore very small numbers of PDDs at these stations. We also already couch our findings in terms of 'significant melting', and note that we only observe this in 2012, which is well known for being an extreme melt year corroborated by other sources of evidence. I think perhaps that the important finding here is actually the degree of discrepancy between the magnitude of extreme events observed/modelled at these locations, and so we have edited this paragraph as follows:

[revised manuscript text omitted]